# hCMV-Mediated Immune Escape Mechanisms Favor Pathogen Growth and Disturb the Immune Privilege of the Eye

**DOI:** 10.3390/ijms20040858

**Published:** 2019-02-16

**Authors:** Katrin Spekker-Bosker, Christoph-Martin Ufermann, Marco Maywald, Albert Zimmermann, Andreas Domröse, Claudia Woite, Walter Däubener, Silvia Kathrin Eller

**Affiliations:** 1Institute of Medical Microbiology and Hospital Hygiene, Heinrich-Heine-University Düsseldorf, 40225 Düsseldorf, Germany; Spekker@uni-duesseldorf.de (K.S.-B.); Christoph-Martin.Ufermann@hhu.de (C.-M.U.); Andreas.Domroese@hhu.de (A.D.); Claudia.Woite@hhu.de (C.W.); daeubene@uni-duesseldorf.de (W.D.); 2Institute of Virology, Heinrich-Heine-University Düsseldorf, 40225 Düsseldorf, Germany; marcomaywald@yahoo.de (M.M.); Albert.Zimmermann@med.uni-duesseldorf.de (A.Z.); 3Bavarian Nordic GmbH, 82152 Martinsried, Germany

**Keywords:** human cytomegalovirus, retinitis, uveitis, chorioretinitis, ocular toxoplasmosis, Indoleamine 2,3-dioxygenase-1, inducible nitric oxide synthase, *Toxoplasma gondii*, human retinal pigment epithelial cells, *Staphylococcus aureus*

## Abstract

Human retinal pigment epithelial (hRPE) cells are important for the establishment and maintenance of the immune privilege of the eye. They function as target cells for human cytomegalovirus (hCMV), but are able to restrict viral replication. hCMV causes opportunistic posterior uveitis such as retinitis and chorioretinitis. Both mainly occur in severely immunocompromised patients and rarely manifest in immunocompetent individuals. In this study, hRPE cells were infected with hCMV in vitro and activated with proinflammatory cytokines. The enzymatic activities of indoleamine 2,3-dioxygenase-1 (IDO1) and inducible nitric oxide synthase (iNOS) were determined. The antimicrobial capacity of both molecules was analyzed in co-infection experiments using *Staphylococcus aureus* (*S. aureus*) and *Toxoplasma*
*gondii* (*T. gondii*), causing uveitis in patients. We show that an hCMV infection of hRPE cells blocks IDO1 and iNOS mediated antimicrobial defense mechanisms necessary for the control of *S. aureus* and *T. gondii*. hCMV also inhibits immune suppressive effector mechanisms in hRPE. The interferon gamma-induced IDO1 dependent immune regulation was severely blocked, as detected by the loss of T cell inhibition. We conclude that an active hCMV infection in the eye might favor the replication of pathogens causing co-infections in immunosuppressed individuals. An hCMV caused blockade of IDO1 might weaken the eye’s immune privilege and favor the development of post-infectious autoimmune uveitis.

## 1. Introduction

Human Cytomegalovirus (hCMV) is an enveloped, double stranded herpesvirus that infects more than 50% of all humans [1,2]. Depending on the efficiency of the host immune system, an hCMV infection is usually asymptomatic but persists lifelong. Human CMV is a species-specific virus and is spread between humans via saliva, blood, and sex. Currently, hCMV is also spread by medical interventions, for example by transfusion of blood products or organ transplants [3].

In addition, hCMV might be spread during pregnancy from mother to child resulting in congenital CMV infections (cCMV), resulting in 5–30 children per 1000 live births being hCMV infected. More than 50% of all cCMV infections take place in newborns from sero-positive mothers while other congenital infections, e.g. rubella or toxoplasmosis, occur only in sero-negative mothers. Up to 90% of cCMV infected individuals will have no clinical manifestations during the newborn period, but about 10–15% of initially asymptomatic children will develop CMV-related disabilities during their childhood. Clinically, sensorineural hearing loss, cognitive impairment, and retinitis are the most frequently diagnosed symptoms in early and late manifested cCMV [4].

In addition, in the pre-antiretroviral era manifested CMV infections were found in up to 70% of human immunodeficiency virus (HIV) infected patients. In most cases, hCMV causes symptomatic eye infections, especially retinitis, but conjunctivitis, keratitis, and uveitis were also observed [5]. Due to modern combinations of antiviral therapies, the frequency of retinitis has declined. However, CMV retinitis still remains the major cause of vision loss in acquired immunodeficiency syndrome (AIDS) patients [6]. Furthermore, CMV retinitis develops in patients after allogeneic haematopoietic stem cell transplantation and hCMV afflicts patients receiving intensive immunosuppressive therapies [7].

In immune competent individuals, monocytes/macrophages act as targets and reservoirs of CMV. Monocytes/macrophages are involved in the control of viral replication [8,9] and trigger T cell proliferation. Cytotoxic effects mediated by natural killer (NK) cells and T cells as well as cytokine-activated blood and tissue cells contribute to the control of virus growth [10]. However, in the eye, normal cellular immune responses directed against pathogens are suppressed since strong proinflammatory reactions might irreversibly damage the non-regenerative neural retina, resulting in blindness [11]. The retinal pigment epithelial (RPE) cells build up a monolayer outside of the neural retina and function as a blood–retina barrier. RPE cells control the transport of nutrients from choroidal capillaries to the neural retina and prevent nonspecific diffusion and transport. In addition, RPE cells can function as antimicrobial effector cells and inhibit the growth of pathogens such as the parasite *Toxoplasma gondii* (*T. gondii*) or the hCMV virus. In the eye, especially, interferon-gamma (IFN-γ) stimulation is necessary to enable RPE cells to control pathogens. Induction of the IFN-γ activated tryptophan-degrading enzyme indoleamine 2,3-dioxygenase (IDO1) is crucial in the defense against *T. gondii* and hCMV [12,13].

A second important immunological characteristic of RPE cells is their ability to establish and maintain the immune privilege in the posterior part of the eye. We and others have described that IDO1 is not only involved in antimicrobial defense but also mediates immunosuppressive effects in tissue cells such as fibroblasts and epithelial cells [14]. Interestingly, hCMV is able to inhibit IDO1-mediated immunoregulatory effects in fibroblast-like cells [15]. Since the kinetics of IDO1 induction of viral growth as well as of hCMV-induced antiviral effect is different in fibroblasts and epithelial cells [16], we analyzed the capacity of hCMV to regulate antimicrobial effects in human RPE cells. In addition, we analyzed the impact of an hCMV infection on the immunosuppressive functions of human RPE cells.

## 2. Results

### 2.1. hCMV Controls Induction of IFN-γ Dependent IDO Expression

Since in vivo CMV elicits an IFN-γ induction in NK cells and T cells which is thereafter released locally, uninfected cells in close proximity are also activated. These cells are simultaneously confronted with infectious virus particles and IFN-γ. Thus, we chose an experimental setting in which we stimulated and infected hRPE cells at the same time point. We have previously shown that hRPE cells can express IDO1 activity and are able to control CMV replication [16]. Here we confirmed these results by flow cytometry (Figure 1). Human RPE cells were stimulated with IFN-γ and infected with hCMV (m.o.i 1) simultaneously. After 24 h we performed FACS analyses and detected the expression of the IFN-γ inducible molecule IDO1 as well as the efficiency of the CMV infection. Human RPE cells stimulated with IFN-γ (200 U/mL) expressed IDO1, detected by increased APC-A levels, whereas no IDO1 was detectable in the unstimulated control group (Figure 1A,B). After a GFP-labelled CMV infection (molecules of infection; m.o.i. 1), more than half of the cells were CMV positive, showing an increased EGFP expression (Figure 1C). Interestingly, virus infected and IFN-γ stimulated cells expressed lower amounts of IDO1 protein than uninfected IFN-γ stimulated cells (Figure 1D).

### 2.2. IDO1-Mediated Antibacterial and Antiparasitic Effects are Lost in hRPE Cells upon CMV Infection

Therefore, we analyzed whether CMV infected hRPE cells maintain their ability to restrict the growth of pathogens, which cause co-infections in the eye such as *S. aureus* or *T. gondii*. As expected, IFN-γ stimulated hRPE cells inhibited the growth of *S. aureus* (Figure 2A). This antibacterial effect was blocked by the IDO inhibitor 1-MT or by supplemental tryptophan (Figure 2A). In contrast, IFN-γ stimulated and CMV infected hRPE cell cultures lost their capacity to restrict the bacterial growth (Figure 2A). In a control group N^G^MMA, the NOS-specific inhibitor was used. No influence on IDO1 function was observed (Figure 2A). In order to validate the potency of the observed antibacterial effect, cfu measurements were performed. The bacterial growth in supernatant of IFN-γ activated hRPE cells was reduced about four orders of magnitude in comparison to unstimulated cells (Figure 2B). This strong antibacterial efficiency was completely blocked upon CMV infection.

Comparable results were obtained in infection studies using the intracellular parasite *T. gondii*. Although unstimulated CMV infected and uninfected hRPE allow the intracellular growth of *T. gondii*, the total parasite proliferation is reduced in CMV infected cells. Therefore, the relative parasite growth, detected in eight independent experiments, is shown as percent of the respective positive control in Figure 2C. IFN-γ stimulated hRPE cells restrict the growth of *T. gondii* (Figure 2C). In contrast, CMV infected cells are no longer able to prevent parasite growth in an IFN-γ dependent manner (Figure 2C).

### 2.3. CMV Infection Results in Lower IDO1 and iNOS Activity in Human RPE Cells

We have previously described that CMV infection and replication in hRPE cells, in comparison to epithelial cells, is somehow slower, while both cell types show a similar kinetic of IFN-γ mediated IDO1 induction [16]. Since the time course of CMV replication is important for the CMV mediated inhibition of IFN-γ signals, we analyzed the effect of CMV on the induction of IDO1 after IFN-γ stimulation. Figure 3A,B indicate that IDO activity, determined by quantification of kynurenine in the supernatants, is reduced in IFN-γ stimulated and CMV infected hRPE cells. The data shows that an multiplicity of infection (MOI) of 1 to 2 is necessary to reach a significant inhibition of IFN-γ induced IDO activity (Figure 3A,B). Since iNOS is also an IFN-γ induced antimicrobial effector molecule in hRPE cells, we determined whether the infection with CMV likewise inhibits iNOS activity. The co-stimulation with IFN-γ, IL-1β and TNF-α resulted in a high production of nitric oxide (NO) (Figure 3C). We found that an MOI of 1 or 2 was necessary to gain a complete block of NO production (Figure 3C). Furthermore, NO production could be reduced by N^G^MMA mediated NOS inhibition (Figure 3C).

### 2.4. iNOS Activity and CMV Infection Impair IDO1-Mediated Immunoregulatory Effects in Human RPE Cells

Next to their antimicrobial effectivity, IDO1 and iNOS are also potent immunoregulatory molecules. Therefore, we characterized the capacity of both enzymes to regulate T cell responses. Human RPE cells were either CMV infected or not and simultaneously stimulated with IFN-γ. After 72 h, peripheral blood lymphocytes were added to hRPE cell cultures and T cell proliferation was induced by OKT3 antibodies. A strong T cell proliferation could be detected in the presence of unstimulated cells, but as expected, the IFN-γ induced IDO1 activity led to a strong reduction of T cell proliferation (Figure 4A). This immunoregulatory effect could be reversed by addition of the IDO-specific inhibitor 1-MT or by tryptophan supplementation (Figure 4A). In contrast, the anti-proliferative effect could not be restored by the addition of N^G^MMA (Figure 4A). In order to analyze if iNOS had an effect on T cell proliferation as well, hRPE cells were stimulated with a cytokine cocktail consisting of IFN-γ, IL-1β, and TNF-α. iNOS positive hRPE cells did not inhibit T cell proliferation (Figure 4B). The blockade of nitric oxide production by N^G^MMA restored the suppressive capacity of hRPE cells, which was clearly IDO1 mediated, since the addition of 1-MT or supplemental tryptophan allowed a full T cell growth (Figure 4B).

Since IDO1 and iNOS activities were inhibited in CMV infected hRPE cell cultures we analyzed the capacity of these cells to mediate immunosuppressive effects. CMV infected hRPE cells lost their capacity to inhibit T cell proliferation after IFN-γ stimulation and IDO1 induction (Figure 4C). This immunoregulation was also maintained in CMV infected, IFN-γ, IL-1β, and TNF-α co-stimulated cells (Figure 4D). This led to the conclusion that CMV had no influence on T cell proliferation even in the presence of nitric oxide.

## 3. Discussion

Toxoplasmosis and hCMV infections have clinical, epidemiological, and immunological similarities. The frequency of infection is high in both cases and about 50% of all humans are sero-positive for hCMV or *T. gondii*, depending on their age. Both pathogens can cause congenital as well as postnatal acquired infections. Clinical manifestations are frequently found in immunosuppressed patients and provoke blindness [4,6,17,18]. *T. gondii* and hCMV infections can result in ocular diseases and both pathogens commonly use RPE cells as target and host cells. Given the high prevalence for both pathogens in humans it is likely that *T. gondii* and hCMV infect the eye and especially hRPE cells simultaneously. Therefore, it makes sense to perform co-infections of hRPE cells with *T. gondii* and hCMV. In prior studies it was described that the IFN-γ induced tryptophan-degrading enzyme IDO1 mediates antiparasitic and antiviral effects in primary human RPE cells [12,13]. The same observation was made in the hRPE cell line ARPE 19 [16].

Here we show that an hCMV infection reduced IDO1 activity in IFN-γ activated cells which subsequently resulted in an inhibition of the IDO1-mediated antiparasitic effect against *T. gondii*. During evolution hCMV has developed several mechanisms to avoid IFN-γ induced antiviral effector mechanisms. For example, hCMV inhibits tyrosine phosphorylation of signal transducer and activator of transcription 1 (STAT1) [19] and decreases Janus Kinase 1 (Jak1) levels [20]. We therefore suggest that these immune escape mechanisms developed by hCMV might favor *T. gondii* proliferation in co-infected RPE cells, which might result in severe ocular toxoplasmosis, especially in HIV patients.

Furthermore, IDO1-mediated tryptophan degradation inhibits the growth of *S. aureus* in hRPE cells. *S. aureus* is the most important pathogen causing bacterial endophthalmitis and it is assumed that *S. aureus* crosses the blood–retina barrier that is build up from endothelial cells. Therefore, we have also co-infected human RPE cells with hCMV and *S. aureus*. We found that IFN-γ activated and hCMV infected hRPE cells were unable to restrict the growth of *S. aureus*. In the clinical setting this interplay might facilitate bacterial endophthalmitis infections in hCMV sero-positive patients. We assume that it is worth to analyze patient’s statistical data hereon in the future.

The defense against several pathogens, especially in murine cells, is mediated by the activation of the inducible nitric oxide synthase (iNOS), which catalyzes the production of nitric oxide (NO) from L-arginine. However, primary hRPE cells were able to produce NO after co-stimulation with IFN-γ and IL-1β. Nevertheless, iNOS induction was not responsible for the IFN-γ induced antiviral and antiparasitic effects. [12,13]. Here we show that iNOS activity was also induced in the ARPE 19 RPE cell line used in this study. We found that the iNOS activity was even more pronounced when hRPE cells were stimulated with a combination of IL-1β and TNF-α. Given the fact that the inhibitory effect of hCMV was not IDO1 but STAT1/JAK1 pathway specific, as discussed above, we showed that hCMV infected ARPE 19 RPE cells lost their capacity to produce NO.

In addition to the discussed antimicrobial effects mediated by hRPE cells, these cells are also endowed with important immunoregulatory capacities. Thus, hRPE cells are important for the immune privilege of the eye. For example, murine RPE cells could present antigen to naïve T cells. However, these T cells were not stimulated and became anergic [21]. In addition, Sugita et al. described a PD1L-PD1 dependent inhibition of T helper 22 cells by human and murine RPE cells [22]. Furthermore, the role of TGF-β production by RPE cells as well as RPE cell-mediated activation of regulatory T cells was reviewed by Keino et al. recently [11].

Since IDO1 and iNOS are both accepted as immunoregulatory factors, we started to analyze the role of both in the regulation of T cell responses by RPE cells and studied the influence of hCMV in this setting. We found that IFN-γ stimulated hRPE cells inhibited human T cell proliferation and this effect was antagonized by the IDO1 inhibitor 1-L-MT or by tryptophan supplementation. Comparable data have been published by Park et al. using a xenobiotic system consisting of hRPE cells and murine T cells [23]. A further hint for IDO1-mediated immunosuppressive effects is the observation that the addition of the IDO1 inhibitor 1-MT increased allogeneic T cell proliferation in the presence of unstimulated hRPE cells [23]. Our finding of IDO1-mediated inhibition of T cell proliferation also fits to the results of a detailed analysis of T cell activation in co-culture with ARPE 19 cells published by Kaestel et al. [24]. They analyzed the activation of T cells in the presence of hRPE cells and observed that the T cells expressed markers of early activation such as enhanced major histocompatibility complex class II antigen expression and increased CD39 expression. However, T cells did not start to proliferate; they showed a lower IL-2 production and a reduced expression of IL-2 receptor alpha and gamma chain and therefore the T cells were arrested in the G1 phase. Such a G1 T cell arrest in the presence of IDO1-positive antigen presenting cells (APC) was also described by Munn et al. In detail, they elucidated that IDO1 induction results in tryptophan starvation and T cells, which have been activated under tryptophan-deficient conditions, were able to synthesize protein, enter the cell cycle, and progress normally through the initial stages of G1, but were unable to enter the G2 phase [25].

In order to analyze the effect of iNOS activity on T cell proliferation, we stimulated hRPE cells with a combination of IFN-γ, IL-1β, and TNF-α which resulted in an optimal iNOS induction in hRPE cells. Surprisingly, we found that cells stimulated with this cytokine cocktail lost their immunosuppressive properties. We have previously published that IFN-γ stimulated human uroepithelial cells expressed IDO1 and inhibited the growth of parasites and bacteria [26]. In addition, RT4 cells expressed iNOS activity after co-stimulation with IFN-γ, IL-1β, and TNF-α. This iNOS activity was unable to mediate antibacterial effects, but it inhibited IDO1-mediated bacteriostasis [27]. In more detailed analyses we could show that NO did not influence IDO1 transcription or mRNA stability. Instead we found an enhanced proteasomal degradation of IDO1 in NO-producing RT4 cells.

Human CMV inhibits IFN-γ mediated antimicrobial effects which might have consequences on the course of additional opportunistic infections, especially on *T. gondii* mediated chorioretinitis. However, the decreased capacity of hRPE to control cellular immune reactions might also be clinically important. Human CMV retinitis is most frequently found in severely immunosuppressed patients. Of course, it is difficult (virtually impossible) to determine the impact of an additional IDO1 deficiency in hRPE during hCMV retinitis. However, hCMV retinitis can also occur in seemingly immune competent patients [28,29]. Here a reduction of the immune privilege of the eye might be important. We suggest that a local IDO1 deficiency might enhance the chance for an unwanted activation of immune responses to unique retinal proteins with a subsequent development of autoimmune uveitis in the presence of an intact cellular immune system.

## 4. Material and Methods

### 4.1. Cells, Media and Reagents

Human retinal pigment epithelial (hRPE) cells ARPE 19 (ATCC, Wesel, Germany), as well as human foreskin fibroblasts (HFF) (ATCC, Wesel, Germany), the *T. gondii* feeder cells, were cultured in Iscove’s modified Dulbecco’s medium (IMDM) (Gibco, Grand Island, NE, USA), supplemented with 5% heat-inactivated fetal calf serum (FCS; Lonza, Verviers, Belgium). Cells were cultured in culture flasks (Costar, Cambridge, MA, USA) in a humidified incubator (37 °C, 10% CO_2_). ARPE 19 cells were passaged weekly in 1:10 ratios using trypsin/EDTA (Gibco). Mycoplasma contamination was regularly excluded by culture methods and PCR. Peripheral blood lymphocytes (PBL) were prepared from heparinised blood of healthy donors after density gradient centrifugation.

### 4.2. Flow Cytometry

hCMV gene expression was controlled using the EGFP-expressing TB40-GFP. Intracellular fluorescence-activated cell scanning (FACS) was performed after fixation and permeabilization using an antibody specific for IDO1 (clone 10.1, Merck, Darmstadt, Germany) detected by an allophycocyanin (APC)-conjugated goat anti-mouse Ig secondary antibody (BD, Heidelberg, Germany). Fluorescence intensities were measured by a FACSCanto II (BD, Heidelberg, Germany) [16].

### 4.3. Stimulation of hRPE Cells

For infection experiments as well as for the determination of IDO1 and iNOS activity, hRPE cells were seeded in 96-well plates (3 × 10^4^ cells per well) and pre-stimulated with recombinant human IFN-γ (0–1000 U/mL) (R&D Systems, Minneapolis, USA) and/or recombinant human IL-1β (100 U/mL) (R&D Systems, Minneapolis, MN, USA), and/or recombinant human TNF-α (100 U/mL) (R&D systems, Minneapolis, MN, USA) for 72 h in a humidified incubator (37 °C, 10% CO_2_). In some experimental groups the NOS inhibitor N^G^monomethyl-l-arginine (N^G^MMA) (Merck, Darmstadt, Germany) (100 µg/mL), IDO inhibitor 1-methyl-tryptophan (1-MT) (Sigma-Aldrich, Deisenhofen, Germany) (1.5 mM), or additional L-tryptophan (Sigma-Aldrich) (100 µg/mL) were used.

### 4.4. Kynurenine Assay

The enzymatic activity of IDO1 directly correlates with the concentration of kynurenine in supernatants of tissue culture cells and the measurement of kynurenine can be used to determine IDO1 activity [30]. The kynurenine content of supernatants from unstimulated or stimulated cells was analyzed using 4-(dimethylamino) benzaldehyde (Ehrlich’s reagent) as described before [30].

### 4.5. Nitric Oxide Assay

Nitrite accumulation in the supernatant of cultured cells was used as an indicator of NO production and was determined by the Griess reaction [31]. In brief, 100 μL cell culture supernatant of stimulated human RPE cells was mixed with 100 μL Griess reagent (0.1% *N*-(1-naphthyl)ethylenediamine in purified water and 2.5% sulfonamide in 15% hydrochloric acid in a 1:1 ratio (Merck, Darmstadt, Germany)). After an incubation time of 15 min, the absorbance was measured at 540 nm (TECAN Sunrise microplate reader, Crailsheim, Germany). The amount of nitrite accumulation was determined using a calibration curve of graded concentrations of sodium nitrite.

### 4.6. CMV Infection

The endotheliotropic strain TB40/E [32] reconstituted from its respective BAC clone TB40/E-BAC4 [33] was obtained from Dr. A. Zimmermann (Institute of Virology, University Düsseldorf, Germany). Purified hCMV stocks were prepared on HFF cells as described before [34]. In brief, infectious supernatants were harvested from HFF cells when they showed a complete cytopathic effect. Virus titers were determined by standard plaque assay in human RPE cells; the titers of the virus stocks were between approximately 3 × 10^6^ and 3 × 10^7^ plaque forming units (p.f.u.) per ml. Infectious supernatants were stored at −80 °C.

Pre-stimulated human RPE cells (3 × 10^4^ cells per well) were infected with 0.01–2 p.f.u. and centrifuged at 800× *g* for 30 min. Then the microtiter plates were incubated in a humidified incubator (37 °C, 10% CO_2_) and the successful infection was observed microscopically by the occurrence of cytopathic effects.

### 4.7. Determination of Toxoplasma Growth

*Toxoplasma gondii* tachyzoites (ME49 strain, ATCC, Wesel, Germany) were maintained in HFF cells in IMDM containing 5% FCS. Extracellular tachyzoites were harvested from culture supernatants by centrifugation, resuspended in tryptophan-free RPMI cell culture medium (Gibco), counted, and used as indicated for infection experiments. After pre-stimulation for 72 h hRPE cells were infected with 2 × 10^4^
*T. gondii* tachyzoites per well. *T. gondii* growth was determined by the ^3^H-uracil incorporation method as previously described [35]. In brief, ^3^H-uracil (0.33 μCi per well) was added 48 h post infection. Cultivation was stopped after additional 24 h by freezing. Parasite growth was determined by measuring incorporated ^3^H-uracil using liquid scintillation spectrometry (1205 Betaplate, PerkinElmer, Jugesheim, Germany).

### 4.8. Determination of Bacterial Growth

A tryptophan–auxotrophic *Staphylococcus aureus* isolate, obtained from a routine diagnostic specimen, was used [36]. *S. aureus* was grown on brain heart infusion agar containing 5% sheep blood (Difco, Hamburg, Germany) at 37 °C in 5% CO_2_-enriched atmosphere overnight. For infection experiments a 24 h old *S. aureus* colony was picked, re-suspended in PBS (Gibco) and serial diluted. Human RPE cell cultures (150 µL) were inoculated with 10 µL of the bacterial dilution containing 10–100 colony forming units (cfu). Cultures were incubated in a humidified incubator (37 °C, 10% CO_2_) overnight. Bacterial growth was monitored by measuring the optical density of re-suspended cultures at 620 nm (TECAN Sunrise microplate reader, Crailsheim, Germany). For the determination of cfu, the infected culture medium was serially diluted in PBS and applied on brain heart infusion agar containing 5% sheep blood in triplicates and incubated overnight. The number of colonies in the respective dilutions was counted.

### 4.9. T Cell Proliferation Assay

Human RPE cells were stimulated as described above and after 72 h 1.5 × 10^5^ freshly isolated Ficoll- separated peripheral blood lymphocytes (PBL)/well were added. PBL were activated using the monoclonal anti-CD3 antibody OKT3, unstimulated PBL served as control group. T cell proliferation was determined after three days by adding ^3^H-thymidine for 24 h. The incorporation of ^3^H-thydimine was detected using liquid scintillation spectrometry (1205 Betaplate, PerkinElmer, Jugesheim, Germany).

### 4.10. Statistical Analysis

In all single experiments replicates (duplicates or triplicates) were measured. The data are presented as mean +/− SEM of a minimum of three independent experiments. For statistical analysis the nonparametric Mann–Whitney U test was used and significant differences were marked with asterisks (* = *p* ≤ 0.05; ** = *p* ≤ 0.001; *** = *p* ≤ 0.0001). The statistical analysis was performed with GraphPad Prism software.

## Figures and Tables

**Figure 1 ijms-20-00858-f001:**
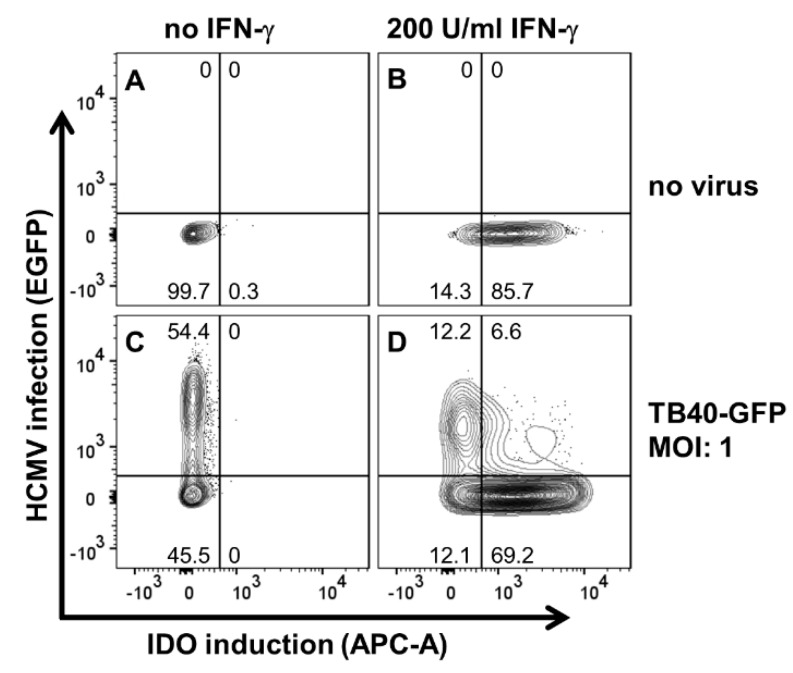
Human cytomegalovirus (hCMV) controls induction of interferon-gamma (IFN-γ) dependent indoleamine 2,3-dioxygenase-1 (IDO) expression. Human retinal pigment epithelial (RPE) cells were either left untreated (**A**) or reated with 200 U/mL IFN-γ (**B**). TB40-GFP (m.o.i. of 1) was used to infect untreated (**C**) or IFN-γ-treated hRPE cells (**D**). Cells were harvested 24 h after infection and/or IFN-γ treatment and stained for expression of IDO1. Enhanced green fluorescent protein (EGFP) expression was used as an infection marker. The numbers in the panels indicate the proportion of IDO1-expressing and/or hCMV-infected cells (% of total population).

**Figure 2 ijms-20-00858-f002:**
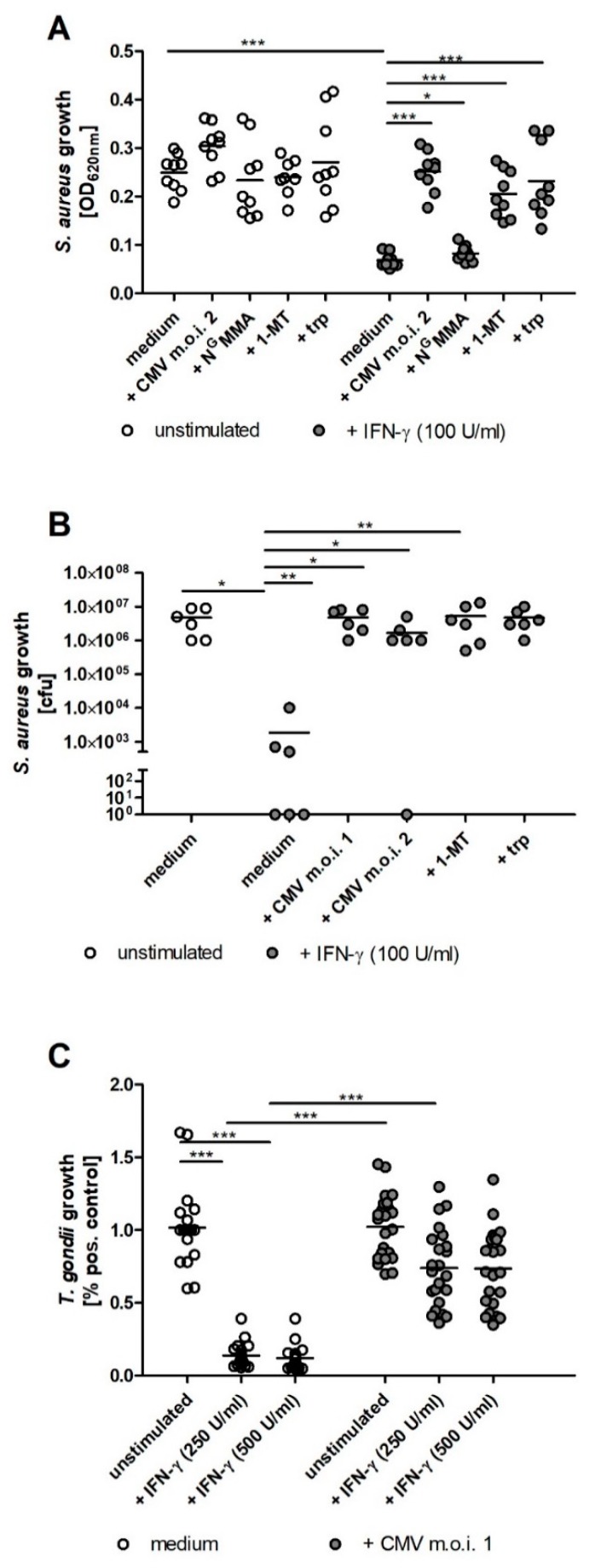
IDO1-mediated antibacterial and antiparasitic effects are lost in hRPE cells upon CMV infection. 3 × 10^4^ hRPE cells were stimulated in 96-well plates with indicated amounts of human IFN-γ (0 to 500 U/mL) in the presence of CMV (m.o.i. 1 or 2), N^G^MMA (100 µg/mL), 1-methyl-tryptophan (1-MT; 1.5 mM), or l-tryptophan (trp; 100 µg/mL). After 72 h the cell culture supernatants were harvested and infected with *Staphylococcus aureus* (10–100 cfu/well) (**A**,**B**) or *Toxoplasma gondii* (3 × 10^4^ ME49 tachyzoites per well) (**C**). *S. aureus* growth was detected by measurement of the optical density at 620 nm after 16 h (**A**) or by counting the colony forming units (cfu; **B**). *T. gondii* growth was measured after three days using the ^3^H-uracil incorporation method. Data are given as mean ± SEM of three (**A**,**B**) or eight (**C**) experiments, each performed in triplicate. Significant differences to the unstimulated or uninfected group were marked with asterisks (n.s. = not significant; * *p* ≤ 0.05; ** *p* ≤ 0.001 and *** *p* ≤ 0.0001). The nonparametric Mann–Whitney U test was used.

**Figure 3 ijms-20-00858-f003:**
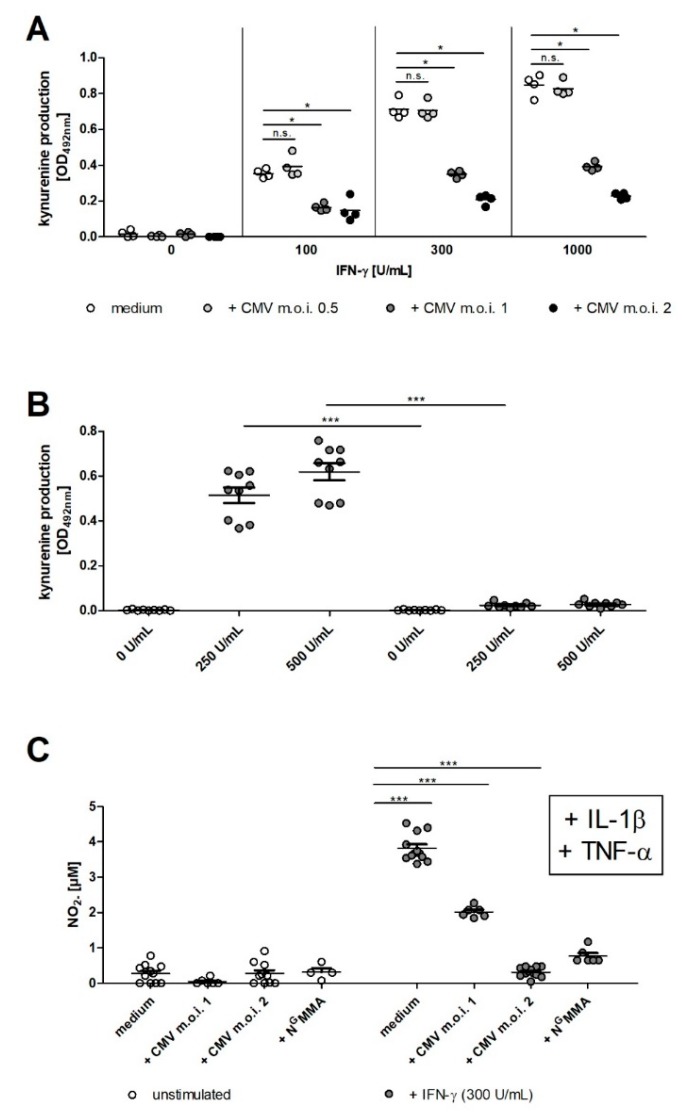
CMV infection results in lower IDO1 and iNOS activity in human RPE cells. 3 × 10^4^ hRPE cells were stimulated in 96-well plates with indicated amounts of human IFN-γ (0 to 1000 U/mL) in the presence of CMV (MOI, 0.05–2) or N^G^MMA (100 µg/mL). After 72 h the cell culture supernatants were harvested and the kynurenine content was determined by optical density measurement at 492 nm, using Ehrlich’s reagent (**A**,**B**) or the nitrite content was quantified by measuring the optical density at 540 nm, using Griess reagent (**C**). Data are given as mean ± SEM of three experiments, each performed in triplicate. Significant differences to the unstimulated or uninfected group were marked with asterisks (n.s. = not significant; * *p* ≤ 0.05; and *** *p* ≤ 0.0001). The nonparametric Mann–Whitney U test was used.

**Figure 4 ijms-20-00858-f004:**
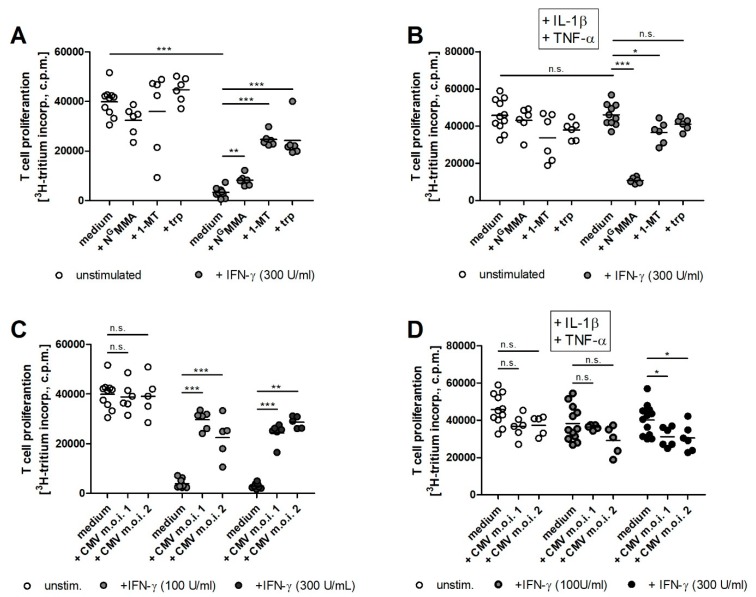
iNOS activity and CMV infection impair IDO1-mediated immunoregulatory effects in human RPE cells. 3 × 10^4^ hRPE cells were stimulated in 96-well plates with indicated amounts of human IFN-γ (0 to 300 U/mL) in the absence (**A**,**C**) or presence of IL-1β and TNF-α (100 U/mL each) and N^G^MMA (100 µg/mL) or 1-methyl-tryptophan (1-MT; 1.5 mM) or l-tryptophan (trp; 100 µg/mL). Simultaneously with the stimulation, the cells were infected with CMV (MOI, 1 or 2) (**C**,**D**) or were left uninfected (**A**,**B**). After 72 h, 1.5 × 10^5^ freshly isolated peripheral blood lymphocytes/well were added. The anti-CD3 antibody OKT3 was used to activate the T cells. T cell proliferation was determined after three days by supplementation of ^3^H-thymidine for 24 h. The incorporation of ^3^H-thydimine was detected as counts per minute (CPM) using liquid scintillation spectrometry. Data are given as mean ± SEM of three experiments, each performed in triplicate. Significant differences to the unstimulated or uninfected group were marked with asterisks (n.s. = not significant; * *p* ≤ 0.05; ** *p* ≤ 0.001 and *** *p* ≤ 0.0001). The nonparametric Mann–Whitney U test was used.

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
