# Peer review of "hCMV-Mediated Immune Escape Mechanisms Favor Pathogen Growth and Disturb the Immune Privilege of the Eye"

_ijms, 2019, doi:10.3390/ijms20040858_

Reviewer 1 Report

In this study human cytomegalovirus (hCMV) was used to infect human retinal pigment epithelial (hRPE) cells and was activated with proinflammatory cytokines. As a next step the 20 enzymatic activities of indoleamine 2,3-dioxygenase-1 (IDO1) and inducible nitric oxide synthase (iNOS) were determined. Also co-infection experiments using Staphylococcus aureus (S. aureus) and Toxoplasma gondii (T. gondii) were used in order to check what happens with the antimicrobial capacity of both enzymes. As explained in details in the Introduction and in the Discussion sections of the manuscript such studies are of high relevance as they can greatly help to clarify the mechanism of eyesight threatening infections in immune suppressed patients. Here it was found that the infection of hRPE cells with hMCV virus blocks IDO1 and iNOS mediated antimicrobial defense. The IFN-γ induced IDO1 dependent immune regulation was severely blocked. Thus the infected hRPE cells were not able to control the infection processes caused by S. aureus and T. gondii as the immune suppressive effector mechanisms in hRPE were strongly inhibited by hMCV.

The Methods are described in sufficient detail and the Results are presented and explained thoroughly and clearly.

 I have just some minor notes:

 - In Methods the point on statistical analysis states that:

  “All experiments were performed in duplicates or triplicates (as indicated in the figure legends) and data are given as mean +/- SEM of a minimum of three independent experiments.”

 It is not immediately evident how the data obtained from experiments done in duplicates can be presented as a mean +/- SEM of a minimum of three independent experiments. Also duplicate experiments are basically to be avoided in experimental studies as they can be misleading. Please clarify the sentence.

 - In the Abstract the abbreviation hRPE also occurs as HRPE. Please correct this.

Author Response

Dear reviewer,

we have revised our manuscript. Here is the list of changes in a point-to-point response:

 Reviewer #1:

1.         In Methods the point on statistical analysis states that:

“All experiments were performed in duplicates or triplicates (as indicated in the independent experiments.”

It is not immediately evident how the data obtained from experiments done in duplicates can be presented as a mean +/- SEM of a minimum of three independent experiments. Also duplicate experiments are basically to be avoided in experimental studies as they can be misleading. Please clarify the sentence.

answer:

Indeed the sentence is misleading. We changed the sentence in the Methods section to: “In all single experiments replicates (duplicates or triplicates) were measured. The data are presented as mean +/- SEM of a minimum of three independent experiments “

2.         In the Abstract the abbreviation hRPE also occurs as HRPE. Please correct this.

answer:

HRPE was corrected to hRPE in the abstract and the correct writing was checked throughout the whole manuscript.

Reviewer 2 Report

The inference may need to be proved by the animal model.

Author Response

Dear reviewer,

we have revised our manuscript. Here is the list of changes in a point-to-point response:

Reviewer #2:

1.         “The inference may need to be proved by the animal model.”

answer:

Indeed an animal model is a good way to proof the inference. However, hCMV shows a strict species specificity and therefore there has been a lack of suitable animal models for a long time. Recently, humanized mouse models in which immune deficient mice are engrafted with human tissues have opened the door for the direct in vivo investigation of viruses with growth restricted to human cells. Unfortunately we do not have proper animals, resources and the permission to perform these animal models.

 We hope that the manuscript is suitable for the publication in the special issue "Host-Cytomegalovirus Interactions: Pro- and Antiviral Key Factors Affecting CMV Infection Outcome or Therapeutic Approaches".

Round  2

Reviewer 2 Report

Hope animal model can be set up one day.